# Clinical Questions and Psychological Change: How Can Artificial Intelligence Support Mental Health Practitioners?

**DOI:** 10.3390/bs14121225

**Published:** 2024-12-19

**Authors:** Luisa Orrù, Marco Cuccarini, Christian Moro, Gian Piero Turchi

**Affiliations:** 1Department of Philosophy, Sociology, Education and Applied Psychology, University of Padova, 35122 Padova, Italy; luisa.orru@phd.unipd.it (L.O.); christian.moro@unipd.it (C.M.); 2Department of Mathematics and Computer Science, University of Perugia, 06123 Perugia, Italy; marco.cuccarini@collaboratori.unipg.it; 3Department of Biology, University of Napoli, 80138 Napoli, Italy

**Keywords:** mental health, clinical questions, clinical psychology, psychological change, machine learning, Artificial Intelligence

## Abstract

Despite their diverse assumptions, clinical psychology approaches share the goal of mental health promotion. The literature highlights their usefulness, but also some issues related to their effectiveness, such as their difficulties in monitoring psychological change. The elective strategy for activating and managing psychological change is the clinical question. But how do different types of questions foster psychological change? This work tries to answer this issue by studying therapist–patient interactions with a ML model for text analysis. The goal was to investigate how psychological change occurs thanks to different types of questions, and to see if the ML model recognized this difference in analyzing patients’ answers to therapists’ clinical questions. The experimental dataset of 14,567 texts was divided based on two different question purposes, splitting answers in two categories: those elicited by questions asking patients to start describing their clinical situation, or those from asking them to detail how they evaluate their situation and mental health condition. The hypothesis that these categories are distinguishable by the model was confirmed by the results, which corroborate the different valences of the questions. These results foreshadow the possibility to train ML and AI models to suggest clinical questions to therapists based on patients’ answers, allowing the increase of clinicians’ knowledge, techniques, and skills.

## 1. Introduction

In the field of clinical psychology, therapists apply various models of intervention, differing (also) on epistemological references and theoretical–methodological perspectives [1,2]. While these latter models serve as orientations in which to operate, sometimes they are sidelined under an individual therapist’s clinical opinion. Indeed, Ref. [3] highlighted that clinical psychologists are driven to use personal theories, despite referring to the same atheoretical manual (DSM-5). From both a theoretical and methodological perspective, this leads to several issues in measuring and treating psychological disorders [4]. A prime example comes from [5], which showed that therapists tend to seek a confirmation of symptoms to validate their diagnosis rather than a disconfirmation. As a result, this could lead therapists to build common-sense theories when observing the patient [6] that may result in negative outcomes [7,8] or even drop-outs from the psychological consultation. Thus, the therapist’s skills and interactive modalities can play a pivotal role in this [9,10]. In addition, another relevant issue underlined by the literature is the difficulty of anticipating and monitoring the progress of psychological change independently from the approach adopted by the therapist [11].

Despite the background and specificity of each theory, all clinical models of intervention are united by the goal of promoting individual, couple, or community mental health, also considering the new forms of clinical manifestations typical of our time [12]. Starting from [13]’s definition, a recent literature contribution highlights how health is defined based on how language is used [14], i.e., by the discursive modalities used to describe it [15]. Thus, mental health can be conceptualized as an interactive and dialogic process [14] based on the use of natural language. From this perspective, psychological change is intended as the modification of the discursive modalities used to configure reasonings and events, contemplating the anticipation of future scenarios from a certain point of a person’s life (linked, for example, to a particular issue or critical event).

Considering the above, clinical psychology needs to know how to observe the interactive–dialogic process in order to improve people’s mental health. The literature shows that there is ample room for the development of therapists’ skill and strategies of intervention.

In the most recent years, methodologies including Machine Learning (ML) techniques have begun to be seen as potentially useful for the field of psychology. Some researchers have already discussed the possible applications of ML in clinical psychology [16] and other more general branches of psychology [17]; the aspect of greatest interest lies in identifying behavioral descriptors useful for predicting or monitoring therapy for specific situations [18,19,20,21]. Shatte et al. [22] identified more than 190 studies that have applied ML for detecting and diagnosing mental disorders, and more than 60 aimed at predicting their progression over time, as well as exploring computerized support for their management [11]. Some applications also employed Artificial Neural Networks (NNs), whose success lies in their ability to learn meaningful embeddings of complex data, such as text [23]; drawing from the field of Natural Language Processing (NLP), these applications involved question answering (related to the personalization and effectiveness of therapies), sentiment analysis, and predicting specific personality traits from patients’ texts [24,25]. Some of the studies reviewed by [22] highlighted some elements of high clinical relevance. For example, Ref. [26] showed that NLP techniques provided accurate results in predicting suicide ideation and psychiatric symptoms in recently dismissed patients; Ref. [27] used unsupervised ML techniques to identify the most predictive psycholinguistic features for successful alcohol abstinence; and Ref. [28] demonstrated benefits in applying supervised ML techniques for matching patients with adequate online support communities. These examples thus corroborate that these innovative techniques can improve the diagnosis and treatment of various clinical conditions [22,26].

In light of this, the current innovation and development of clinical psychology raises some questions: how can we observe and monitor psychological changes in clinical settings? What are the elements that therapists use to conduct clinical consultations that foster psychological change? How do these elements differ from each other? How can ML and Artificial Intelligence (AI) actually support therapists? This work attempts to take up these questions, and proposes a methodology and specific research experimentation based on the analysis of the interactive–dialogic process. We describe the methodology and the related research conducted with the support of ML, so as to develop a useful model and insights to direct clinical consultations and promote psychological change.

### Theoretical–Methodological References and Scope

According to the studies on clinical models of intervention, what patients recount as “illness or distress” and what they and their therapist share in the consultation pathway can only be generated using language. Indeed, natural language is what humans use to give sense to the reality in which they find themselves living [29]. What is peculiar in a clinical setting is the possibility for the therapist to observe the impact of patients’ stories on their mental health. This is crucial for promoting psychological change, i.e., modifying people’s storytelling related to relevant issues and mental health difficulties. In other words, this can transform how a condition and/or a problem is represented and foster other narratives that open up different coping strategies.

Some studies report that there is still no agreement on which are the factors that lead to change [30,31], and the elements that therapists use to direct psychological consultations—therapy goals, treatment principles, treatment techniques, and the therapy relationship [31,32]. However, an element that is certainly common between clinical settings are the (therapist) questions. Questions are the “engine” that activates and sustains the process of psychological change. Indeed, as argued by [15,33], the patient’s narrative does not exist a priori, but is produced in the interaction with the therapist through his/her questions [6,34,35,36]. The question serves to investigate the patient’s needs and requests and to direct his/her narrative toward scenarios of healthy living which are possible but not yet practiced.

In order to explore how the use of natural language can promote psychological change, we chose Dialogic Science as our theoretical–epistemological approach [6,34,35,36], applying its methodology: MADIT (Methodology for the Analysis of Computerized Text Data). MADIT can be included within the Computational Text Analysis Methods (CTAM [37,38]), but differs from the most widespread methods. Indeed, in the most recent years, these have focused on the syntactic and semantic features of language, exploiting content and sentiment from text [39,40]; thus, they consider the meaning of the text alone. Adding to this level, MADIT studies the discursive process of sense-generation, i.e., how humans shape the reality of sense they live in [40]. The observation of such a process is performed through MADIT’s own “denomination” process, which differs from a categorization of content/meaning. In fact, it does not seek an answer to the question, “what is the text about?”. Instead, the question becomes, “what purpose does the text serve?”. Although currently leading to a higher error rate than categorization, applying this question allows the distinctive tracing of the elements that characterize each specific text/narrative, which may be different even when the contents/meanings are the same (see also [40]).

In the field of clinical psychology, MADIT allows the study and analysis of how natural language is used in daily—and thus clinical—interactions [34], generating peculiar realities of sense (of mental health). These latter are called discursive configurations [41]. Thus, MADIT provides a more in-depth level of analysis, going beyond the syntactic–semantic elements of patients’ narratives (which are still considered) and focusing on the sense-generation architecture of them [34,40]. Recent studies [34,36,40,42] have led researchers to formalize 24 language use modalities, named Discursive Repertories (DRs). DRs are categorized into three typologies [34]:Generative DRs promote or keep pressure towards the management of critical situations and the creation of new realities of sense;Stabilization DRs concur to keep unchanged the ways that outline the ongoing reality of sense;Hybrid DRs develop in both a generative or stabilization orientation based on the DR they link with.

According to Dialogic Science assumptions, the ways patients answer clinical questions represent a part of how they shape their realities of sense. Specifically, they represent what they use to pragmatically guide themselves in everyday life to deal with problems, difficulties, etc. Therefore, the formulation of a clinical question stands as the privileged tool that allows a therapist to observe and orient the patient’s discursive configuration within the clinical setting. Applying MADIT for questions’ formulation enables the therapist to define what interactive process underlies the question itself, and to anticipate its impact for psychological change. Therapists’ focus should lie in the purpose of the question and in the language modality used for its formulation. The purpose of the question is what steers patients’ processes of sense-generation, placing their discursive configuration within specific coordinates. Based on MADIT, four types of questions have been defined, distinguished by their purpose:Description: they pursue the objective of detecting the discursive configuration, gathering what is brought by the patient;Evaluation: they pursue the objective of deepening the text offered by the patient, bringing in other elements and specifying them;Generating change: they pursue the objective of shifting the discursive configuration of the patient toward healthier scenarios and intervening following that direction;Maintenance: they pursue the goal of consolidating what has been built and developed during the clinical intervention with the patient up until that specific moment.

Given the potential and actual impact of questions for mental health promotion in clinical settings, ML can come into play as a valuable support for therapists. For example, ML techniques could be used to investigate the relationship between questions and answers, providing useful data on which questions produce a greater shift in patients’ narratives. Or again, generative AI models trained on a clinical questions–answers dataset could suggest the therapist different question choices, to be selected by him/her (in anticipation) for different phases of a psychological consultation. Luxton [43] in particular already stated that AI-enabled VR human avatars have the potential to be implemented for several ‘person-to-person’ interactions in mental healthcare, including psychological treatments, assessments, and testing.

The aim of this paper is to describe our experimentation on MADIT’s ML model learning process in understanding clinical questions’ purpose. MADIT uses ML as a support tool to increase the accuracy in observing patients’ narratives, through an ever-increasing rigor of analysis and the definition of its error rate. In particular, the ML model follows MADIT’s distinctiveness of focusing on the more in-depth level of analysis related to the language modalities used by people for sense-generation. Compared to other ML approaches and models, it adds a layer of analysis on the already available ones, such as syntactical and semantic (which are the current state-of-the-art). Our aim was to explore the capacity of our ML model to understand such an in-depth level of analysis, as a currently unexplored task that we consider useful for the field’s literature. In fact, it could add to the results already achieved in content analysis, where it is known that ML models perform well (in tasks with a limited number of categories).

For this experiment specifically, we observed how description and evaluation questions are used by MADIT’s ML model and if there were differences in the resulting analysis of patients’ answers. The experiment investigated the implications of transferring the question purpose to the NN in the process of denominating DRs. These observations may allow further considerations regarding the impact of clinical questions for psychological consultations, as well as new ones related to their degree of “predictivity” for mental health promotion.

## 2. Materials and Methods

The experimental dataset processed by MADIT’s ML model consisted of a total of 14,567 text excerpts. These were gathered from verbatim transcripts of questions and answers between therapists and patients in psychological consultation. It is a heterogeneous sample of 28 people, aged 18 to 65, including both males and females. The clinical approaches adopted were interactionist and dialogical, and verbatim transcription of interviews started in 2006. The excerpts consist of 303,316 graphic forms, for a total of 14,507 DRs. All text are in the Italian language.

The excerpts were selected taking into consideration the purpose of the question that produced the answers and their corresponding DRs. For the processing phase, the dataset was organized based on the type of question.

For evaluating the accuracy of the DRs’ denomination, we employed the following metrics for all ML models tested (see also [44]):Precision: the number of correctly labeled items out of all items that were labeled (correctly or not with that class of DR);Recall: the number of correctly labeled items compared to the total number of items that belong to that DR class;F1-score: the harmonic mean between precision and recall;Accuracy: the number of correct predictions out of all predictions made.

The tested models include Bidirectional Encoder Representations from Transformers (BERT), which analyzes the text from left to right and vice versa. Here we present only the results obtained with it, due to the better output compared to other models.

Without going into the technical details of the model (see Table 1 for its hyperparameters), the process of automated text denomination through MADIT is modelled as a pipeline of two subtasks computed simultaneously: (1) text segmentation; and (2) prediction of DRs for a single text span.

As per (1), text segmentation is the task of splitting text into meaningful segments. Starting from raw text, for the MADIT methodology these segments correspond to a sequence of DRs. Thus, each text span is defined as a segment of text in which the DR does not change. The ML algorithm for text segmentation is trained to detect the boundaries between text spans that represent a change of DR.

As per (2), DR prediction is formalized as a text classification task: from a text span derived from task 1) (optimal segmentation of the text), the algorithm predicts a label from a finite set. The label corresponds to a specific DR, taken from the 24 DRs represented in the table of Discursive Repertories.

MADIT’s text analysis methodology comprises 6 steps (shown in Figure 1).

The translation of these steps into a ML model for automated text analysis was accomplished through the following algorithm–machine (see also Figure 2):Encode question and answer in the input;Use what has already been administered in the training phase of the ML algorithm to encode the argumentative ‘joints’ (the part(s) of text that could represent the DR) defined by the question (and/or its DR), vocabulary, position of graphic forms, presence of particles and, more generally, relevant features in congruence with what has been administered;Denominate the most plausible DR for the response text, based on the coding obtained.

**Figure 2 behavsci-14-01225-f002:**
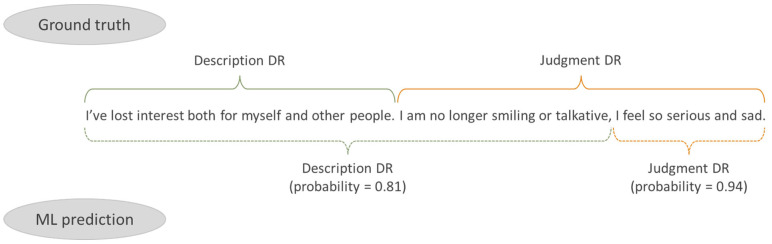
Example of text information encoding, with ground truth (senior human analyst—see below) and ML prediction.

The starting assumptions in transferring and using the question’s purpose were as follows:By changing the question, the trajectory of the discursive configuration changes and, in turn, the possibilities of psychological change increase;Some DRs may be more likely to occur than others;With description questions, a wider range of DRs may be more likely to occur;With evaluation questions, a shorter range toward hybrid DRs may be more likely to occur.

To confirm these hypotheses, we compared text denomination across ML models in two different experimental situations, where the following were provided: (1) only the text of the patients’ answers, without the question’s purpose; (2) the question’s purpose and the text of the answers. The question’s purpose was passed to the model in two steps: transmitting the entire question string, and then transmitting the information on the question type based on its purpose. The structure and processes of the two experimental cases are represented in Figure 3.

Preliminary tests made it possible to highlight how, in not using the question as a methodological step, the process witnessed is not of knowledge construction (or sense-generation), but of content recognition and classification; it is not possible to denominate DRs correctly. Therefore, even in the application of ML, if the question is not fully part of the algorithm there will be only a process of classificatory (and content) recognition of DRs. Extending this issue to the clinical setting, it is as if the therapist focused only on the term “anxiety” and not on how the patient used that construct specifically in relation to his/her personal situation. Conversely, the therapist should look at the patient’s discursive modalities, observing how he/she answers questions configuring a peculiar reality of sense. As reference, novice psychologists using MADIT in the clinical setting—defined as Junior—reach a 66% accuracy in identifying and denominating patients’ discursive modalities (DRs); differently, psychologists well-established in the methodology reach 99.9% accuracy [40]. Given the above, the following questions guided our research: how much does focusing on the purpose of the question impact on the ways to foster psychological change? Can our ML model recognize the difference in question purpose by denominating answers differently? It is clear to us that if we could define which questions are more effective, we could anticipate the direction of psychological change itself—the mental health impact of the psychological consultation—and thus help maintain or generate it.

## 3. Results

### 3.1. Experimental Case 1: No Question vs. Question String

Concerning the first experimental case, from Table 2 it is possible to see how the absence of the question string affects the ML model’s classification performance: we present the values of precision, recall, and F1-score for each DR and the overall accuracy for trial A (presence of question string) and trial B (absence of question string). Bold values indicate the highest value between the trials A and B.

Without the question string, the values of precision tend to decrease even in a very substantial way, with some exceptions (the DRs of Justification, Confirmation, Anticipation, Possibility, and Prescription). The DRs that most clearly represent this decrease are Non Answer (−0.24 in F1), Prevision (−0.12 in Rec.), Proposal (−0.22 in F1), Reshaping (−0.25 in Prec., −0.16 in Rec., −0.32 in F1) and Certify Reality (−0.1 in F1). An extreme case is represented by the DR of Consideration, for which the values pass from 0.09 (Prec.), 0.31 (Rec.), and 0.14 (F1) with the use of the question to 0 without its use. The performance variations, which in the two tests ranged from 0.01 (a value also attributable to randomness affecting the NN training) to 0.1 (not attributable to randomness), indicate that the mere presence of the question constitutes an element of denomination change for some DRs. In previous tests, both the full text and the reference DR of the question had not performed incrementally in the model. Therefore, informing the model about the presence of the question brings a clear advantage in the quality of denomination: for almost all DRs the precision, recall, and F1-score values increase, as does the overall accuracy. However, it affects much less than it does in the case of a human; in fact, human analysts raise their accuracy degree from 0.51 (non-use of the question) to 0.66 (using the question).

This finding allows the development of the transmission of the question’s purpose information to the model.

### 3.2. Experimental Case 2: Adding Information About the Question

Considering the results from experimental case 1, we provided the dataset for ML analysis with the additional information regarding the purpose of the question (description or evaluation) and analyzed the distribution of DRs emerging from these two types of questions. Our underlying thoughts were regarding what specific information contained in the question was used by humans to produce a denomination that differs when faced with different questions. The original dataset was adjusted by splitting the answers to description questions (asking patients to start describing their clinical situation—12,737 texts) and evaluation questions (asking patients to detail how they evaluate their situation and mental health condition—324 texts). Data were normalized for subsequent comparison. Table 3 presents the DRs predicted by MADIT’s ML model, split by question type and in order of frequency.

As it can be noted, the first part of the distribution (the first nine DRs) is nearly identical, and only Evaluation and Specification are switched. The differences between the two groups of questions occur in the central and last part of the distribution, where DRs begin to occur in a different order of frequency. The DRs of Anticipation, Reshaping, and Confirmation emerge only in the group of answers to description questions. Furthermore, two meaningful differences regarding the DRs of Opinion and Comment can be noted: for evaluation questions, their values range within the maximally present DRs (within 0.08 probability), whereas for description questions they surpass this threshold and lie as the less incident DRs.

These empirical results thus seem to confirm the different valences of the two questions at the theoretical level, and in turn the hypotheses initially formulated (see Section 2). Providing the ML model with the information on the purpose of the question does indeed show different profiles of answers. First of all, a wider range of DRs occurs in response to description questions. In addition, apart from the first part of the distribution, for description questions, stabilization and hybrid DRs occur more frequently, while for evaluation questions, hybrid DRs are more likely and generative ones appear earlier. This output is also consistent from a theoretical point of view; in fact, in a psychological consultation, it is first useful to collect the patient’s problem in detail, thus maximizing the variety of discursive modalities that can be used through description questions. Afterwards, the therapist asks the patient to elaborate on the problem, making him/her formulate evaluations and articulate thoughts perhaps not previously expressed, which already orient the narrative toward a change onset (through generative DRs).

## 4. Discussion and Conclusions

The experimental cases developed to investigate the impact of questions’ purpose on changing patients’ discursive configurations allow us to highlight some considerations. First, it can be noted how the presence of the question improves the accuracy of text denomination. In particular, knowing that a question exists and reading it through the referred DR is useful information for MADIT’s ML model, though not as prominent as for humans [29]. Therefore, we wondered what information human intelligence was using that had not yet been offered to the ML model to increase its denomination accuracy. The second experimental case allowed us to validate the theoretical assertion according to questions with different purposes generating different answers. In fact, a different distribution of DRs was obtained based on questions with different purposes: only for description questions are present the DRs of Anticipation, Reshaping, and Confirmation. For evaluation questions, instead, there is a higher probability of finding the DRs of Possibility, Opinion, and Comment.

In addition, it is possible to assert that the ML model’s accuracy increases thanks to the transfer of the information on the questions’ purpose. This increase highlights that MADIT’s ML algorithm refers to the interactive–dialogic process instead of the contents exclusively [29,42]. Thus, in the future development of tools leveraging ML algorithms for actually supporting practitioners in question formulation, it will be essential to consider and further precise the question recognition step, specifically linking it to its purpose.

These results could lead to future insights useful for the scientific community and experts in the field. In particular, these first experimental data could contribute to developing the research and the interventions considering and anticipating the most effective ways to improve peoples’ mental health [29]. This is relevant because it allows the underlining of specific elements useful for psychological consultations. Among these are the relevance of questions to direct patients’ journey towards psychological change; the importance of considering the processual dimension of the question (its purpose), and not only the contents (with what words it is formulated); and the possibility to use and develop a theoretical and methodological model enabling therapists to anticipate and measure the impact of their interventions on patients’ critical narratives.

Aligning with the aim described in the Introduction, with these conclusive suggestions, we advocate a shift for clinical interventions, where therapists start considering mental health as an interactive and dialogic process (and not only a “state of being free from illness or injury”) [14,40,45] and coherently conduct their psychological consultations.

Our experimental research comes with some limitations, first of all, the number of question considered (two of four types) and the size of the dataset. Also, considering the heavily exploratory and preliminary nature of our research, we did not perform statistical analysis and compared only the adopted metrics. Thus, future developments include strengthening the experiments and studying the frequency distributions to address the significance of the differences between results. This would allow specific hypothesis testing to compare cases, increasing the values on the calculated metrics, and reducing the error rate of the model. Another limitation of our study is that, due to time and resources, we were not able to test all the available ML models (especially the newer ones); indeed, we stuck to BERT for its more than acceptable results for such a task. However, we are planning to expand the range of models to explore, and we have already started with Llama. In addition, two other research lines are currently active: one is aimed at integrating different datasets to analyze and contemplate the ‘generating change’ and ‘maintenance’ purposes of clinical questions; the other is experimenting with our model for the production of synthetic answers to different clinical questions, in order to observe even more precisely how models grasp the differences between them and produce varying responses. About the latter, some preliminary tests already show that a generative AI model trained on MADIT’s ML model is able to predict which are the most likely DRs of an answer that a question can generate. Further developments of our work include the launch of some other experimental tests in applied research. On one hand, we plan to explore datasets related to specific and current psychological issues—such as eco-anxiety [12] and social anxiety [41] (to mention a few)—comparing results and exploring similarities and differences of possible clinical usefulness. On the other hand, we also intend to focus on similar thematic areas which are not strictly clinical: network and organization cohesion, persuasion, civic and social mediation, and legal consultations.

We believe that a strength of our experimental research lies in the degree of innovation, inasmuch as it is a research line specifically aimed at grounding future applications in the clinical field, thus being usable within and outside the scientific community, up to the occupational field. Aligning with this, another distinctive feature of our research is the introduction of a new and more in-depth level of text analysis. The application of MADIT provides, in fact, a type of data accounting not only for which problems patients bring in their psychological consultation journey, but especially for how they make sense of these problems within their narrative: which discursive modalities they adopt and what the repercussions and the clinical impact are of using one or another.

In conclusion, we think that there is a premise for ML techniques and, even more, for generative AI models, to actually support therapists in the future (with some already available results [26,27,28]). However, it is equally true that at the current state-of-the-art, the accuracy levels of these models are not sufficient enough for the type of job therapists do. Therapists are distinctly more skilled and precise, which is required considering the substantial degree of responsibility they have toward their patients. Therefore, on one hand it is certainly necessary to increase the amount of data for training, in order to enhance models’ precision. On the other hand, it could be useful to deploy these innovative supports in the psychological field step-by-step, thinking about specific and targeted applications. For example, (i) transcribing and highlighting patients’ most used discursive modalities, to facilitate therapists’ post-hoc work of observation–reasoning; (ii) anticipating the development trajectory of the psychological consultation, suggesting clinical questions that cross the most relevant contents and critical discursive modalities of patients’ narratives; and (iii) simulating therapist–patient interactions, with the aim of teaching and training current and future therapists and increasing their knowledge, techniques, and skills.

## Figures and Tables

**Figure 1 behavsci-14-01225-f001:**
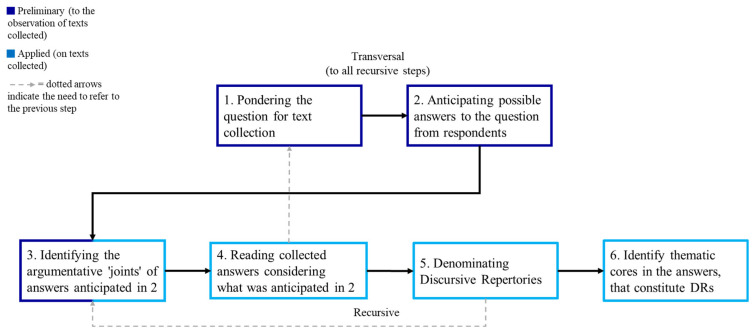
Steps for denominating text (algorithm).

**Figure 3 behavsci-14-01225-f003:**
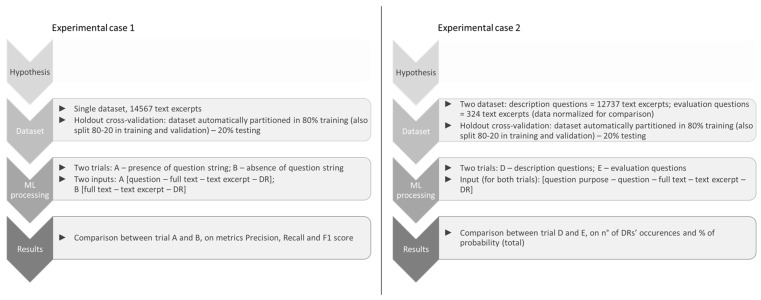
Structure and processes of the two experimental cases.

**Table 1 behavsci-14-01225-t001:** BERT model structure.

Model	Pretrained Weights	Batch Size	Learning Rate	Freeze To
BertClfier	bert-base-italian-xxl-uncased	16	1e -05	1
AdamW Eps	Max Epochs	Patience Epochs	Embed Dim	Activation Function
0.0001	200	20	768	ReLU

**Table 2 behavsci-14-01225-t002:** MADIT’s ML model classification performance, comparing (A) presence of the question string and (B) absence of the question string.

Discursive Repertoires	PrecA	PrecB	RecA	RecB	F1A	F1B
Anticipation	0.06	**0.07**	**0.13**	0.07	**0.08**	0.07
Cause of Action	**0.40**	0.32	**0.48**	0.44	**0.43**	0.37
Comment	**0.26**	0.20	**0.22**	0.20	**0.23**	0.20
Confirmation	0.25	**0.27**	**0.35**	0.31	**0.29**	**0.29**
Consideration	**0.09**	0.00	**0.31**	0.00	**0.14**	0.00
Contraposition	**0.48**	0.43	**0.64**	0.63	**0.55**	0.51
Description	**0.68**	0.64	**0.46**	0.45	**0.55**	0.53
Declaration of Aims	**0.30**	0.26	**0.60**	0.53	**0.40**	0.35
Generalization	**0.23**	0.16	**0.17**	0.11	**0.19**	0.13
Judgment	**0.63**	0.56	**0.52**	**0.52**	**0.57**	0.54
Justification	0.27	**0.28**	**0.42**	0.36	**0.33**	0.31
Implications	**0.31**	0.30	**0.41**	0.34	**0.35**	0.32
Non Answer	**0.42**	0.38	**0.62**	0.61	**0.71**	0.47
Opinion	**0.52**	0.46	**0.67**	**0.67**	**0.59**	0.54
Possibility	**0.40**	**0.40**	**0.43**	0.39	**0.42**	0.39
Prescription	**0.37**	**0.37**	**0.74**	0.67	**0.49**	0.48
Prevision	**0.26**	0.24	**0.52**	0.40	**0.34**	0.30
Proposal	**0.28**	0.23	0.52	**0.58**	**0.36**	0.14
Reshaping	**0.31**	0.06	**0.57**	0.14	**0.40**	0.08
Targeting	**0.37**	0.29	0.51	**0.57**	**0.43**	0.39
Certify Reality	**0.55**	0.50	**0.25**	0.16	**0.34**	0.24
Specification	**0.33**	0.30	0.39	0.32	0.36	0.31
Evaluation	0.29	**0.30**	**0.34**	**0.34**	0.31	**0.32**
Accuracy	-	-	-	-	**0.43**	0.39
Macro avg	**0.35**	0.30	**0.45**	0.39	**0.38**	0.32
Weighted avg	**0.47**	0.43	**0.43**	0.39	**0.43**	0.38

**Table 3 behavsci-14-01225-t003:** Discursive Repertoires’ distribution for description and evaluation questions.

RDs–Description Question	*n*° of Occurrences	% Probability	DRs—Evaluation Question	*n*° of Occurrence	% Probability
Certify Reality	2213	0.17			
Judgment	2173	0.17			
Description	1885	0.15	Certify Reality	75	0.2
Opinion	687	0.05	Judgment	69	0.18
Comment	606	0.05	Description	52	0.14
Evaluation	579	0.05	Opinion	44	0.12
Specification	538	0.04	Comment	31	0.08
Generalization	504	0.04	Specification	16	0.04
Contraposition	406	0.03	Evaluation	13	0.03
Implications	391	0.03	Generalization	12	0.03
Justification	343	0.03	Contraposition	11	0.03
Cause of Action	315	0.02	Prescription	11	0.03
Prevision	278	0.02	Possibility	7	0.02
Possibility	269	0.02	Proposal	6	0.02
Declaration of Aims	267	0.02	Justification	5	0.01
Non Answer	251	0.02	Cause of Action	4	0.01
Prescription	208	0.02	Prevision	4	0.01
Confirmation	186	0.01	Implications	3	0.01
Targeting	173	0.01	Non Answer	3	0.01
Proposal	96	0.01	Consideration	2	0.01
Anticipation	47	0.003	Declaration of Aims	2	0.01
Consideration	41	0.002	Targeting	2	0.01
Reshaping	2	0.0001			

## Data Availability

The data presented in this study are available on request from the corresponding author due to privacy reasons.

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
