# Peer review of "Clinical Questions and Psychological Change: How Can Artificial Intelligence Support Mental Health Practitioners?"

_behavsci, 2024, doi:10.3390/bs14121225_

Round 1
Reviewer 1 Report
Comments and Suggestions for Authors
This is a very interesting experiment. It seems thoughtfully designed and worth consideration. To be more useful, I would suggest these changes:
1. The English needs extensive editing. There are numerous grammatical and word choice errors.
2. The tables need more definition and explanation connected to the content.
3. The narrative does not flow well. This may be partly due to the language issues, but it also needs editing for flow of content.
4. I would include more background on how practitioners make these decisions. That seems to be a key to honing the accuracy of the ML protocol.
5. Please include a more extensive statement on next steps with this protocol.
The English desperately needs to be edited, and until that is done, I am not sure I can provide more specific feedback. The piece does not flow well, and the grammatical errors require the reader to make assumptions about their actual meaning. I think the really important part of this work, that they could certainly highlight more clearly, is whether or not machine learning can enhance clinicians' work with patients. It appears to me that expert clinicians are far better at this process than AI at this point. I think the usefulness at some point may be in assisting the experts by making decisions more quickly, and could also assist in training new clinicians, but not until it is as accurate as expert humans. Comments on the Quality of English Language
This article needs significant editing before publication for an English speaking audience.
Author Response
First, we would like to thank the reviewer for the feedback and precious suggestions. Here follows the point-by-point response to the comments:
1. The English needs extensive editing. There are numerous grammatical and word choice errors.
We thank the reviewer for pointing out that English needed to be adjusted. It is now clear to us that the previous version of the paper could only be understood by us, but not by an external reader from a linguistic standpoint. We have now made several changes throughout the paper, trying to simplify both the concepts and the sentences, aligning words and avoiding errors. Any other linguistic suggestion will certainly be welcomed.
2. The tables need more definition and explanation connected to the content.
First of all, we specified the tables’ captions. We also introduced the results’ tables with some more details and expanded their explanation in relation to the hypothesis and relevant contents.
3. The narrative does not flow well. This may be partly due to the language issues, but it also needs editing for flow of content.
Considering the linguistic editing, we simplified multiple sentences and concepts, in order to achieve a better and more clear flow of our narrative.
4. I would include more background on how practitioners make these decisions. That seems to be a key to honing the accuracy of the ML protocol.
Following the suggestion, we added some more elements and references related to this issue in the Introduction, paragraph 1.1 (although, as reported, it is still debated and not exhaustive).
5. Please include a more extensive statement on next steps with this protocol.
Following the limitations of our study, in the Conclusions section we extended the current and future research steps with our experimentations and ML model.
The English desperately needs to be edited, and until that is done, I am not sure I can provide more specific feedback. The piece does not flow well, and the grammatical errors require the reader to make assumptions about their actual meaning. I think the really important part of this work, that they could certainly highlight more clearly, is whether or not machine learning can enhance clinicians' work with patients. It appears to me that expert clinicians are far better at this process than AI at this point. I think the usefulness at some point may be in assisting the experts by making decisions more quickly, and could also assist in training new clinicians, but not until it is as accurate as expert humans.
We highly thank the reviewer for the suggestion. In the last part of the Discussion and Conclusions section we added an entirely new part with our thoughts on the topic reported.
Reviewer 2 Report
Comments and Suggestions for Authors
Well prepared and presented. This paper will enrich research in the same field. I recommend that the author provide a more detailed explanation of the clinical significance of integrating Artificial Intelligence with specific psychological treatment modalities in the introduction. Additionally, it would be helpful to discuss the clinical implications of utilizing these techniques. Lastly, the author should address both the strengths and limitations of the study.
Author Response
First, we would like to thank the reviewer for the feedback and precious suggestions. Here follows the point-by-point response to the comments:
I recommend that the author provide a more detailed explanation of the clinical significance of integrating Artificial Intelligence with specific psychological treatment modalities in the introduction. Additionally, it would be helpful to discuss the clinical implications of utilizing these techniques.
We added some more relevant elements and references in two parts of the Introduction, related both to the clinical relevance of ML-AI and to the some linked clinical benefits.
Lastly, the author should address both the strengths and limitations of the study.
We further specified the limitations of our study, adding two more, and added the elements we believe are the strengths and distinctive features of our research.
Reviewer 3 Report
Comments and Suggestions for Authors
The study aims to the application of AI models to assist mental health practitioners, focusing on their capacity to learn from various types of questions and assess the influence of question types on eliciting patients' responses. Here are my comments:
1. The research background is not clearly introduced. There is a lack of a clear presentation on the role and progress of the text analysis method in MADIT, making it difficult for readers outside the field to understand.
2. The research purpose is unclear. What is the significance of this machine learning method compared to other methods, such as statistical-based or other machine learning approaches? Or, what new information can be obtained by adopting this machine learning method?
3. The research process is not clearly described. There is a lack of detailed descriptions of key aspects such as the specific implementation process, data partitioning, and cross-validation methods.
4. The specific implementation of the two experiments is vague. In the experimental design, for the "absent question condition", no clear problem information is input. So what information is actually input? Additionally, the encoding of problem information could be further detailed or illustrated with examples.
5. The results are not clearly described. In Experiment 1, the authors mentioned "minor effect" between the conditions with and without questions, was statistical analysis performed to assess the differences? And how should the results of Experiment 2 be interpreted?
Comments on the Quality of English Language1. The language and grammar of this manuscript is very poor and needs further improvement.
2. The format of the references is confusing.
Author Response
First, we would like to thank the reviewer for the feedback and precious suggestions. Here follows the point-by-point response to the comments:
1. The research background is not clearly introduced. There is a lack of a clear presentation on the role and progress of the text analysis method in MADIT, making it difficult for readers outside the field to understand.
We thank the reviewer for this comment, allowing us to introduce more elements on the topic from the literature. Regarding MADIT specifically, we extended its founding elements, adding to the section 1.1. In that part we also cited another our contribution, which specifically describe MADIT’s background and progresses: due to this (and in order not to repeat in the same way that reference) the added text is a brief summary of the highlights of that contribution.
2. The research purpose is unclear. What is the significance of this machine learning method compared to other methods, such as statistical-based or other machine learning approaches? Or, what new information can be obtained by adopting this machine learning method?
Thanks to the reviewer for pointing out this issue. We tried to answer these questions by adding some more details where we present MADIT, as well as where we mention the related ML model. Also, we extended the research purpose to make it more explicit, hoping this improve clarity and its understanding.
3. The research process is not clearly described. There is a lack of detailed descriptions of key aspects such as the specific implementation process, data partitioning, and cross-validation methods.
We tried to increase the clarity of the key aspect of our research process in a graphical form, providing the information mentioned, as well as the inputs for each experimental case.
4. The specific implementation of the two experiments is vague. In the experimental design, for the "absent questioncondition", no clear problem information is input. So what information is actually input? Additionally, the encoding of problem information could be further detailed or illustrated with examples.
We added a new figure explaining the encoding of text information. We specify that, coherently with other argumentations of the paper and the journal subject area, we provided examples and figures that we think will be understood by other psychologist (thus being not totally detailed from a technical- computer science perspective).
5. The results are not clearly described. In Experiment 1,the authors mentioned "minor effect"between the conditions with and without questions, was statistical analysis performed to assess the differences? And how should the results of Experiment 2 be interpreted?
Regarding both experimental cases, we did not perform statistical analysis, and considered as sufficient (for now) the comparison of the adopted metrics. Our reasoning for not conducting further statical analysis links to the heavily exploratory and preliminary nature of our research; however, we recognize their relevance and, in turn, added this issue as a current limitation and future development of our research. Moreover, in order not to give rise to misunderstandings, we have replaced terms that may refer to statistical analysis and added other parameters for comparison (e.g., Junior and Senior accuracy values). We have also expanded the results’ interpretation from experimental case 2, adding information and references to our research hypothesis.
- The language and grammar of this manuscript is very poor and needs further improvement.
We reread the paper in its entirely, and it is now clear to us that its previous version could only be understood by us, but not by an external reader from a linguistic standpoint. We have now made several changes, trying to simplify both the concepts and the sentences, aligning words and avoiding errors. Any other linguistic suggestion will certainly be welcomed.
- The format of the references is confusing
We have checked and now adjusted all references following MDPI style.
Round 2
Reviewer 1 Report
Comments and Suggestions for Authors
This is very much improved. I believe it may now contribute significant new information to the field. Thank you for your thoughtful and thorough responses to the review suggestions.
Comments on the Quality of English LanguageOnce you have a complete draft not showing the changes, I would suggest one more very close read to catch some minor errors that are difficult to see in the marked up version.
Author Response
This is very much improved. I believe it may now contribute significant new information to the field. Thank you for your thoughtful and thorough responses to the review suggestions.
We would like to thank again the reviewer for all the suggestions provided, which allowed us to improve the quality of the paper.
Once you have a complete draft not showing the changes, I would suggest one more very close read to catch some minor errors that are difficult to see in the marked up version.
We re-read the paper and identified some errors and some unaligned words, that we proceeded to correct.
Reviewer 3 Report
Comments and Suggestions for Authors
The article is much better. I have no other issues.
Comments on the Quality of English LanguageAttention should still be paid to the expression of some terms, such as "machine learning" and "ML".
Author Response
The article is much better. I have no other issues.
We would like to thank again the reviewer for all the suggestions provided, which allowed us to improve the quality of the paper.
Attention should still be paid to the expression of some terms, such as "machine learning" and "ML".
We re-read the paper and identified some errors and some unaligned words (such as the one mentioned), that we proceeded to correct.